# Extending the Toolkit for Beauty: Differential Co-Expression of *DROOPING LEAF*-like and Class B MADS-box Genes during *Phalaenopsis* Flower Development

**DOI:** 10.3390/ijms22137025

**Published:** 2021-06-29

**Authors:** Francesca Lucibelli, Maria Carmen Valoroso, Günter Theißen, Susanne Nolden, Mariana Mondragon-Palomino, Serena Aceto

**Affiliations:** 1Department of Biology, University of Naples Federico II, 80126 Napoli, Italy; francesca.lucibelli@unina.it (F.L.); mariacarmen.valoroso@unina.it (M.C.V.); 2Matthias Schleiden Institute of Genetics, Friedrich Schiller University Jena, 07743 Jena, Germany; guenter.theissen@uni-jena.de (G.T.); nolden_susanne@yahoo.de (S.N.); 3Department of Cell Biology and Plant Biochemistry, University of Regensburg, 93040 Regensburg, Germany

**Keywords:** *DROOPING LEAF*, flower development, gene expression, Orchidaceae, YABBY transcription factors

## Abstract

The molecular basis of orchid flower development is accomplished through a specific regulatory program in which the class B MADS-box *AP3/DEF* genes play a central role. In particular, the differential expression of four class B *AP3*/*DEF* genes is responsible for specification of organ identities in the orchid perianth. Other MADS-box genes (*AGL6* and *SEP*-like) enrich the molecular program underpinning the orchid perianth development, resulting in the expansion of the original “orchid code” in an even more complex gene regulatory network. To identify candidates that could interact with the *AP3/DEF* genes in orchids, we conducted an in silico differential expression analysis in wild-type and peloric *Phalaenopsis*. The results suggest that a YABBY *DL*-like gene could be involved in the molecular program leading to the development of the orchid perianth, particularly the labellum. Two YABBY *DL/CRC* homologs are present in the genome of *Phalaenopsis equestris*, *PeDL1* and *PeDL2*, and both express two alternative isoforms. Quantitative real-time PCR analyses revealed that both genes are expressed in column and ovary. In addition, *PeDL2* is more strongly expressed the labellum than in the other tepals of wild-type flowers. This pattern is similar to that of the *AP3/DEF* genes *PeMADS3/4* and opposite to that of *PeMADS2/5*. In peloric mutant *Phalaenopsis,* where labellum-like structures substitute the lateral inner tepals, *PeDL2* is expressed at similar levels of the *PeMADS2-5* genes, suggesting the involvement of *PeDL2* in the development of the labellum, together with the *PeMADS2-PeMADS5* genes. Although the yeast two-hybrid analysis did not reveal the ability of PeDL2 to bind the PeMADS2-PeMADS5 proteins directly, the existence of regulatory interactions is suggested by the presence of CArG-boxes and other MADS-box transcription factor binding sites within the putative promoter of the orchid *DL2* gene.

## 1. Introduction

The Orchidaceae is one of the widely distributed and most diversified families of angiosperms. Their evolutionary success is possibly due to sundry causes such as epiphytism, extraordinary adaptive capacities to different habitats, highly specialized pollination strategies, and diversified flower morphology [1,2,3]. Despite the diversity of flower colors, sizes, shapes, and appendages, the floral organs of orchids share a common organization (Figure 1). There are three outer tepals in the first floral whorl; in the second whorl, the three tepals are distinguished into two lateral inner tepals and a median inner tepal called lip or labellum. This organ often has a peculiar morphology and bears distinct color patterns (Figure 1a,b). Female and male reproductive organs are fused to form the gynostemium or column, at whose apex are located the pollinia. The ovary is placed at the base of the gynostemium, and its development is activated by pollination [4].

The labellum is a central organ in orchid pollination because of its strikingly distinct morphology and its direct opposition to the gynostemium. Therefore, its showy color patterns and structures are visual attractants and it act as a landing platform that guides pollinators towards the gynostemium. Because the labellum is the uppermost perianth organ, its role in pollination depends on becoming the lowermost through resupination, a 180° developmental rotation of the flower pedicel or ovary (Figure 1a) [5].

Bilateral symmetry or zygomorphy in orchids is a syndrome defined by the association of several characters (e.g., labellum and the developmental suppression of adaxial stamens). This association took place early in Orchidaceae evolution and became the basis for the progressive addition of further innovations like pollinaria, a spur, or showy markings on the labellum [4]. The concurrence of these floral features is considered a key morphological innovation in the two most derived and diverse orchid subfamilies Epidendroideae and Orchidoideae [4]. Together they mediate the specialized relationships of this family with pollinators, facilitating the processes of prezygotic reproductive isolation [6,7].

Because of the central role of the labellum in orchid reproduction, its developmental origin is a subject of intense study [4,8,9,10]. In the last decade, several gene regulatory models inspired by the more general angiosperm ABC model [11,12] helped to explain the developmental specification of the distinct orchid perianth organs [13,14,15,16,17,18]. Specifically, the “orchid code” argues that the diversification of the organs of the orchid perianth is due to the combined differential expression of class B MADS-box genes belonging to the *AP3/DEF* group [13,14,15]. The “homeotic orchid tepals” (HOT model) proposes a combinatorial action of homeotic MADS-box proteins consistent with the “orchid code” [16]. The more recent “P-code” model hypothesizes a pivotal role of the class B and *AGL6* MADS-box genes in forming the orchid perianth [19].

In order to understand the more extensive regulatory network behind orchid flower development, we and others have found that, like *AP3/DEF*s, also candidate *SEP*-, *FUL*-, *AG*-, and *STK*-like MADS-box genes have been duplicated in the Orchidaceae. However, only some of them are differentially expressed in association with the distinct flower organs. For instance, in developing *Phalaenopsis* flowers, we observed that *SEP3*-like and *DEF*-like genes have common expression domains. This shared domain of expression suggests that both candidates are associated with labellum specification, and that similar positional cues determine their expression domains [20]. Elucidating the nature of the positional cues behind the development of specific orchid flower organs is a central question to understand the developmental program of this family.

Top candidates for providing the positional information for differentially expressed MADS-box genes are CYCLOIDEA-like (CYC-like) transcription factors [13], which are well known for their role in flower bilateral symmetry specification in core eudicots [21,22,23,24,25]. Comparative studies of *CYC*-like genes identified several major, well-supported monocot-specific clades and reported the first *CYC*-like genes in orchid species [26,27,28]. Additional studies also showed that the DDR regulatory module composed of the MYB factors DIVARICATA, DRIF, and RADIALIS, responsible in *Antirrhinum majus* for bilateral flower symmetry [29,30], seems to be conserved in orchids [31,32,33]. However, the critical CYC-like transcription factor that activates the transcription of *RADIALIS* in *A. majus* [21,34] is not conserved in orchids. Moreover, the current literature reports contrasting results [26,27,28,35], possibly because the functional equivalent of *CYC*, if it exists, has not yet been identified in orchids.

Our interest in identifying additional components of the regulatory network determining orchid flower organ identity prompted us to conduct a preliminary in silico differential expression study using RNA-seq data of *Phalaenopsis*. This preliminary work suggests a scenario where MADS-box genes and members of the plant-specific family of transcription factors termed YABBY contribute to labellum development.

During the course of angiosperm evolution, the YABBY *DROOPING LEAF/CRABS CLAW* (*DL/CRC*) genes came to regulate the development of different structures like the carpel, nectaries, or the leaf mid-rib [36]. In addition, *DL/CRC* and other members of the YABBY gene family like *FILAMENTOUS FLOWER* (*FIL*) [37,38] respectively determine flower meristem and organ identity in *Arabidopsis* and rice [37,38]. In the rice flower meristem, the expression domain of *DL* is delimited by the class B MADS-box gene *SUPERWOMAN1* [37,39], thus suggesting a regulatory relationship between them. Additional evidence of regulatory interaction between *DL/CRC* and MADS-box genes comes from maize. In this species, the co-orthologs *drl1* and *drl2* have a potential antagonistic relationship with *silky1*, the ortholog of the class B *APETALA3/DEFICIENS* gene, during floral patterning and establishment of floral bilateral symmetry [40].

The existence of a regulatory relationship between *DL/CRC* and MADS-box genes in model dicot and monocot species inspired us to explore the role of *DL*-like genes in orchids. In this family, gene duplication and differential expression of *DEF*-like class B MADS-box genes play a pivotal role in modularizing the perianth [13,14,15,16,17,18,19,41,42]. However, it is not yet clear as to which positional cues determine their expression domains, resulting in flower bilateral symmetry. The present study tests the hypothesis that *DL*-like orchid genes are associated with the development of distinct orchid flower organs. To this purpose, we compared their patterns of expression with those of *DEFICIENS*-like MADS-box genes *PeMADS2, PeMADS3, PeMAD4,* and *PeMADS5* (*PeMADS2-PeMADS5*) in wild-type and peloric *Phalaenopsis* flowers. These mutants have labellum-like structures that substitute the lateral inner tepals, thus lacking the bilateral symmetry of the perianth, and are especially useful to study genes possibly involved in orchid perianth formation. Next, we tested whether the co-expression of *DL*-like and *DEF*-like genes also involves direct protein–protein interactions via yeast two-hybrid assays. Finally, we scanned the putative promoters of the *DL*-like genes of *Phalaenopsis* and *Dendrobium* to identify conserved motifs with possible regulatory functions.

## 2. Results

### 2.1. Identification of Transcription Factors Differentially Expressed in the Labellum

Our initial RNA-seq screening of the *Phalaenopsis* hyb. “Athens” (Figure 1c,d) inner-perianth transcriptome showed over 78% of the read pairs mapped to the *Phalaenopsis equestris* genome v 1.0 [43]. About 68% of the transcripts annotated (21,200 genes) are expressed in the flower organs analyzed with at least 1 TPM (transcripts per kilobase million). Labellum-like lateral inner tepals of peloric flowers and wild-type labella share 98% of all expressed genes. This indicates that these organs express almost the same genes, strongly suggesting that they have the same organ identity (Appendix A).

Analyses of differential gene expression yielded an interesting group of transcripts significantly up- or downregulated in wild-type lateral inner tepals compared to the labellum (Appendix A). Among them, we identified transcripts that are possibly associated with labellum development, encoding DROOPING LEAF-like proteins (DL-like) and the class B MADS-domain protein PeMADS2 (Appendix A). In our analysis, two DL-like transcripts are downregulated in wild-type lateral inner tepals, in comparison to their wild-type and peloric labella levels. Transcripts of class B MADS-box gene PeMADS2 are upregulated in wild-type lateral inner tepals, just as documented by qPCR in the “orchid code” [13,14,15]. Furthermore, *CYC-TB1*-like genes are expressed in lateral inner tepals and labellum at levels under 1 TPM. This extremely low level of expression of *CYC*-*TB1*-like genes during orchid development has also been observed in previous studies [26].

We then conducted an in silico differential expression analysis using publicly available reads of the perianth organs of wild-type and peloric mutant *Phalaenopsis* hyb. “Brother Spring Dancer” KHM190 [44]. We mapped and quantified the reads against the transcriptome of *Phalaenopsis* hyb. “Brother Spring Dancer” assembled from the Illumina raw reads. In this case we also found transcripts encoding class B MADS-box proteins differentially expressed among the organs of the wild-type plant, and detected differential expression for a transcript encoding a DL-like protein (Appendix A). In particular, in the wild-type *Phalaenopsis* this *DL*-like transcript showed a 3 to 4 log_2_ FC expression in labellum than in lateral inner tepals. No significant difference was observed between the transcripts of this gene in the labellum and labellum-like lateral inner tepals of the peloric mutant *Phalaenopsis* hyb. “Brother Spring Dancer” (Appendix A).

The differential pattern of expression of the *DL*-like transcript is analogous to those observed in MADS-box *DEF*-like genes *PeMADS3* and *PeMADS4*, which are highly expressed in the wild-type labellum and labellum-like structures of peloric mutants [14,45]. This similarity suggests an association between the activity of *DL*-like and *DEF*-like homeotic genes and the development of the labellum.

Further in silico analyses of the reference transcriptome of *Phalaenopsis* hyb. “Brother Spring Dancer” identified two *DL*-like transcripts, *PeDL1* and *PeDL2*, each with two different isoforms.

We confirmed the presence of these transcripts by the PCR amplification of cDNA from perianth tissues of *Phalaenopsis* hyb. “Athens” followed by cloning and sequencing, and deposited the sequences in GenBank with the accession numbers MW574592, MW574593 (*PeDL1_1* and *PeDL1_2*), MW574594, and MW574595 (*PeDL2_1* and *PeDL2_2*). The longest isoforms of both transcripts (*PeDL1_1* and *PeDL2_1*) encode proteins containing a C2C2 zinc-finger domain at the N-terminus and a YABBY domain, whereas both the alternative isoforms encode proteins missing the C2C2 zinc-finger domain completely (*PeDL1*_*2*) or partially (*PeDL2*_*2*) (Appendix A). The PeDL1_1 (189 aa) and PeDL2_1 (196 aa) proteins are 64.3% similar, with highly conserved YABBY domains and more variable C2C2 zinc-finger domains. In comparison, the region spanning from the C2C2 to the YABBY domain and the C-terminal region are the less-conserved parts of these proteins (Appendix A).

### 2.2. Genomic Organization of the PeDL1 and PeDL2 Genes

Reconstruction of the genomic organization of the *PeDL1* and *PeDL2* genes based on BLAST analyses of the longest *PeDL* transcripts against the assembled genome of *Phalaenopsis equestris* [43] showed the *PeDL* genes have seven exons and six introns (Figure 2). The large intron 4 is particularly rich in repetitive sequences. This feature has affected the correct assembly of the *PeDL1* and *PeDL2* genes, which were both split in two different genomic scaffolds (Scaffold000404_23 and Scaffold000404_21 for *PeDL1*; Scaffold000061_46 and Scaffold000061_45 for *PeDL2*).

The alignment of the short transcripts *PeDL1_2* and *PeDL2_2* with the corresponding genomic region revealed the presence of a putative alternative transcription start site within intron 1 of *PeDL2* and intron 2 of *PeDL1*, resulting in transcripts whose ATG start codon is located within exon 2 and exon 3, respectively (Figure 2).

### 2.3. Differential Expression of the PeDL1 and PeDL2 Genes

To analyze the expression pattern of *PeDL1* and *PeDL2* in the floral organs of *Phalaenopsis*, we performed quantitative real-time PCR on cDNA from different organs of the wild-type *Phalaenopsis* hyb. “Athens” dissected from floral buds of ~1 cm (B2 stage, Figure 1c). Both genes are highly expressed in the column and ovary. However, the *PeDL2* isoforms are also highly expressed in the labellum relative to outer and the other inner tepals. These results confirm the initial in silico differential expression analysis (Figure 3).

Then, to verify the conservation of these expression patterns and follow them along with flower development, we examined the expression profile of *PeDL1* and *PeDL2* in the perianth tissues of *P. aphrodite* at different developmental stages (Figure 1a,b). As shown in Figure 4, all but *PeDL2_1* have low expression levels in all the perianth organs (outer tepals, inner tepals, and labellum) from the earliest stage B1 to OF (open flower). Interestingly, the isoform *PeDL2_1* is expressed at high levels in the labellum during the first developmental stages. Its expression decreases over time, with a statistically significant negative correlation between expression level and stage (Spearman correlation *r* = −1, *p* = 0.0028).

To test the hypothesis that *PeDL2* is associated with the development of distinct perianth organs, we analyzed the expression pattern of the isoforms *PeDL2_1* and *PeDL2_2* in two *Phalaenopsis* peloric mutants bearing labellum-like structures in place of lateral inner tepals. The peloric *Phalaenopsis* hyb. “Athens” shows an increased expression of both *PeDL2* isoforms in the labellum-like structures compared to the lateral inner tepals of the wild-type (Figure 5). In particular, the mean difference of the expression between lateral inner tepals and labellum decreases from −2.71 (wild-type) to −1.93 (peloric) for *PeDL2_1* and from −4.82 (wild-type) to −0.83 (peloric) for *PeDL2_2*. In the peloric *Phalaenopsis* hyb. “Joy Fairy Tale” there are no significant differences found in the expression levels of *PeDL2* in the inner and outer perianth organs (Figure 6). Additionally, no significant differences were detected in the expression of *PeDL1_1* and *PeDL1_2* in the perianth of wild-type and both peloric *Phalaenopsis* mutants (Appendix A).

### 2.4. Differential Expression of the PeMADS2-PeMADS5 Genes

To compare the expression profile of the *DEF*-like genes in the perianth organs of wild-type and peloric *Phalaenopsis*, we performed real-time PCR experiments on cDNA of wild-type and peloric *Phalaenopsis* hyb. “Athens” (Figure 7) and of the peloric *Phalaenopsis* hyb. “Joy Fairy Tale” (Figure 8). As expected, *PeMADS2* and *PeMADS5* are less expressed in labellum than in outer and inner tepals in the wild-type *Phalaenopsis*. Genes *PeMADS3* and *PeMADS4* show an opposite behavior, being more expressed in the labellum than in other organs of the wild-type perianth. In the peloric *Phalaenopsis* “Athens”, the mean difference between the expression levels of the *PeMADS2-PeMADS5* genes in labellum-like structures and labellum decreases due to the reduced (for *PeMADS2* and *PeMADS5*) or the increased (for *PeMADS3* and *PeMADS4*) expression in the labellum-like structures (Figure 7).

In the peloric *Phalaenopsis* hyb. “Joy Fairy Tale”, the differences in expression level between the labellum-like structures and lip are not significant, except for *PeMADS4*, which shows a higher expression in the labellum-like structures than in labellum (Figure 8).

### 2.5. Protein Interaction: Y2H Analysis

We used the yeast two-hybrid (Y2H) assay to determine if the proteins PeMADS2-PeMADS5 and PeDL2_1 can interact (Appendix A). Our results show that the DEF-like proteins of *Phalaenopsis* do not directly interact with PeDL2_1. We also checked the ability of PeDL2_1 to bind the GLO protein PeMADS6, equally expressed in all the perianth organs [14], also revealing the absence of direct interaction (Appendix A). In addition, we verified the ability of both the isoforms of PeDL1 and PeDL2 to interact with each other, showing the absence of direct interaction in all the possible combinations (Appendix A). As a positive control of the Y2H experiments, we tested the ability of PeMADS2-PeMADS5 to interact with PeMADS6. The results confirm that PeMADS6 can interact with each of the DEF-like proteins of *Phalaenopsis*, although with different strengths, as previously reported (Appendix A) [46].

### 2.6. Conserved Regulatory Motifs

To search for conserved motifs within the promoters of the *PeDL* genes, we analyzed the 3000 bp upstream of the translation start site of the *DL2* genes of *Phalaenopsis equestris* (*PeDL2*) and *Dendrobium catenatum* (*DcDL2*). The MEME analysis revealed motifs shared by the putative promoters of *PeDL2* and *DcDL2* (Figure 9). Two motifs (Motifs 1 and 3) have a relatively well-conserved position within the ~300 bp upstream of the translation start site. These motifs were not found when the analysis was repeated using the shuffled sequences of the putative promoters (Appendix A) and are not present within the putative promoter of the *DL1* gene (Appendix A).

The TOMTOM analysis of Motif 1 against the JASPAR Core Plants database shows that it contains a putative binding site for a TCP protein. The same analysis conducted on Motif 3 revealed that it contains a putative binding site for an SBP-type zinc-finger protein (Figure 9).

The search of known transcription factor binding sites (TFBSs) within the putative promoters of the *DL2* genes of *Phalaenopsis* and *Dendrobium* through PLANTPAN 3.0 [47] identified putative conserved elements belonging to different transcription factor families. For example, in addition to the TCP and SBP binding sites, AP2/ERF, MYB/SANT, and MADS-box binding sites (CArG-boxes) were identified.

The specific search of CArG-boxes gave positive results for the variants CC(A/T)_7_G and C(A/T)_8_G. In particular, one CC(A/T)_7_G site is present in both the *PeDL2* and *DcDL2* putative promoters. In addition, four and six C(A/T)_8_G sites are located within the *PeDL2* and *DcDL2* promoters, respectively (Figure 9). One variant CC(A/T)_7_G and four C(A/T)_8_G CArG-boxes are also present within the putative promoter of *DcDL1*.

## 3. Discussion

Flower formation is the outcome of a complex developmental program in which environmental and genetic factors cooperate. The genetic pathway that drives the correct formation of the floral organs and the establishment of floral symmetry has been studied in detail in model species, where some transcription factor families play a relevant role, mainly MADS-box [11,12], TCP [21], MYB [29], and YABBY [38]. In orchids, the morphology of the flower organs and the establishment of bilateral floral symmetry have been widely studied, resulting in orchid-specific regulatory models where the coordinated action of MADS-box genes explains the formation of the orchid outer, lateral inner tepals, and labellum [13,14,15,16,17,18,19]. In the perspective of a broader, integrated view of these models, recent studies have suggested a possible involvement of TCP [26,27,28,35,48] and MYB [31,33] transcription factors in the developmental program leading to the formation of the orchid perianth, in particular of the labellum. In contrast, the possible involvement of the YABBY transcription factors in this developmental process is still unexplored. Based on these premises and the existence of a regulatory interaction between the YABBY transcription factor DL/CRC and the class B MADS domain transcription factors in rice (OsMADS16) [37,39] and maize (silky) [40], we tested the hypothesis of a similar regulatory relationship in orchids during the formation of the perianth organs, in particular of the labellum.

### 3.1. Paralogous DL-Like Genes in Orchidaceae

Our results support the identification of two *DL*-like genes in the genome of *P. equestris*: *PeDL1* and *PeDL2* [49]. These genes belong to the CRABS CLAW/DROOPING LEAF clade. Each of them is part of one of the sister clades resulting from an Orchidaceae-specific duplication early after the divergence of subfamilies Apostasioideae and Vanilloideae (Appendix A). Our results agree with the finding that *PeDL1* and *PeDL2* are expressed in the column and ovary of *Phalaenopsis* (Figure 3) [49]. This expression profile suggests that like in *Oryza*, *Zea*, *Triticum*, *Sorghum,* and *Arabidopsis*, *PeDL1* and *PeDL2* are also involved in carpel development [49,50,51].

### 3.2. Different Transcripts of DL-Like Genes

We found two differentially spliced transcripts of the *PeDL1* and *PeDL2* genes of *Phalaenopsis*, differing at the 5’ terminus (Figure 2) and encoding proteins completely (PeDL1_2) or partially (PeDL2_2) missing the C2C2 zinc-finger domain (Appendix A). Although we scanned the transcriptomes of various orchid species present in the orchid-specific database Orchidstra 2.0 [52] and OrchidBase 2.0 [53], we did not find similar alternative short transcripts of the *DL* homolog genes. Our initial in silico identification of the *PeDL1_2* and *PeDL2_2* isoforms was verified by PCR, sequencing, and real-time PCR experiments using isoform-specific primers. Our results confirmed the existence of differentially spliced isoforms for both *PeDL* genes. The failure to find alternative transcripts of *DL*s in other orchids might be due to the kind of transcriptomes deposited in the orchid-specific database. This data generally represents transcripts of the whole inflorescence, with possible under-representation of isoforms expressed specifically in few types of cells or organs. Outside orchids, we found the annotation of two isoforms of both the DL genes of *Zea mays drl1* (https://maizegdb.org/gene_center/gene/GRMZM2G088309, access date 18 January 2021) and *drl2* (https://www.maizegdb.org/gene_center/gene/GRMZM2G102218, access date 18 January 2021). In particular, the predicted alternative isoform of the *drl2* gene encodes a short protein missing the C2C2 zinc-finger domain, as in *Phalaenopsis*. Unfortunately, functional or expression data for the *drl* isoforms are not available, and their role is still unknown. Further analyses are needed to assess the function of the truncated isoforms that might work as competitive inhibitors and thus have a regulatory function.

In *Arabidopsis*, YABBY proteins form homo and heterodimers [54]. In particular, the CRC protein forms homodimers and can interact with the YABBY protein INO [55]. In contrast to CRC, our results indicate that PeDL1, PeDL2, and their short isoforms, form neither homo- nor heterodimers (Appendix A), showing that the ability of CRC/DL proteins to homo- and heterodimerize is not conserved among plants. This unexpected result is in agreement with that reported in a recent study on the *DL*-like genes of *Phalaenopsis* [56] and might be due to sequence divergence after duplication, resulting in the loss of the ability to form homo- and heterodimers.

### 3.3. Divergent Patterns of Expression of PeDL1 and PeDL2 during Flower Development

Interestingly, the expression of the two *PeDL* genes in the perianth organs of wild-type *Phalaenopsis* is not overlapping. In contrast to very low expression levels of *PeDL1* in all perianth organs from early to late floral buds, *PeDL2* has a higher level of expression in the labellum than in outer and lateral inner tepals. This trend decreases steadily towards anthesis (Figure 3). Considering that the expression of *DL* in *O. sativa* is restricted to the flower meristem and developing carpels, the expression of *PeDL2* in the perianth is unusual for a *DL*-like gene, and is the first evidence of a possible novel regulatory function acquired by these genes after duplication early in orchid evolution. Our hypothesis of the recruitment of *PeDL2* in orchid perianth development is supported by the expression pattern of the gene in orchid peloric mutants where the inner tepals are substituted by labellum-like structures. In the peloric *Phalaenopsis* “Athens” (Figure 1d), early expression of both *PeDL2* isoforms increases in labellum-like inner lateral tepals compared to the wild type (Figure 5). In addition, the peloric *Phalaenopsis* “Joy Fairy Tale” (Figure 1e) shows similar expression of *PeDL2* in labellum-like structures and labellum (Figure 6). These results support the relationship between the combinatory expression of *PeDL2*, *PeMADS3,* and *PeMADS4* transcripts and labellum development.

### 3.4. The “Orchid Code” beyond MADS

The idea of an “orchid code” enriched by the function of *PeDL2* during the labellum development fully fits with the regulatory profile of the other well-known components of this model: the *DEF*-like MADS-box genes *PeMADS2-PeMADS5*. In wild-type *Phalaenopsis*, the expression in the perianth of *PeDL2* has a similar pattern in the labellum and lateral outer tepals as *PeMADS3* and *PeMADS4* and is opposite to that of *PeMADS2* and *PeMADS5*.

The transcription patterns of *PeDL2* and *PeMADS2-PeMADS5* in wild-type and peloric *Phalaenopsis* allow us to suggest that during the formation of the labellum there could be regulatory interactions between PeMADS2-PeMADS5 and PeDL2, based on different possible molecular mechanisms: either PeMADS proteins bind to regulatory DNA of the *PeDL2* gene (i.e., protein–DNA interactions), or PeMADS and PeDL2 proteins interact (protein–protein interactions) (Figure 10). Although our results from the Y2H analysis do not reveal the ability of PeDL2 to bind any of the PeMADS2-PeMADS5 proteins, a direct protein–protein interaction cannot be definitely excluded, as it could require the formation of a multimeric protein complex.

Alternatively, the regulatory interaction between PeDL2 and PeMADS2-PeMADS5 might be carried out at the transcriptional level, with PeMADS2/5 functioning as transcriptional repressors or PeMADS3/4 as transcriptional activators of the *PeDL2* gene. The MADS-box proteins are transcription factors that bind conserved sites on DNA with the consensus sequence CC(A/T)_6_GG, the canonical CArG-box, or its variants [57]. The presence of the variant CArG-boxes CC(A/T)_7_G and C(A/T)_8_G (Figure 9), known transcription factor binding sites of MADS-domain proteins, in multiple sites of the putative promoter of *DL2* of *Phalaenopsis* and *Dendrobium* suggests that orchid DEF-like proteins or other MIKC-MADS domain genes could regulate the transcription of *DL2*. The presence of multiple CArG-boxes even strongly suggests that tetrameric complexes of MIKC-type proteins (“floral quartets”) are constituted [19,58]. In addition, the presence of shared transcription factor binding sites of TCP and SBP proteins, conserved in sequence and spatial organization in the putative promoters of *DL2*, suggests that other transcription factors could also modulate the expression of this gene to the expression domains here documented.

### 3.5. Conclusions

The molecular basis of orchid flower development is only partially understood. The main components of the orchid “toolkit for beauty” are MADS-box transcription factors; however, other transcription factor families (TCP and MYB) contribute to the differentiation of the organs of the orchid perianth. Our study proposes further expanding this complex developmental program, including the YABBY *PeDL2* of *Phalaenopsis* among the genes responsible for the labellum differentiation. Future studies should be focused on understanding the way of interaction among the different players of this fascinating developmental program to shed light on their regulatory connections.

## 4. Materials and Methods

### 4.1. Plant Material

The orchids used in this study were the wild-type *Phalaenopsis* hyb. “Athens” and *P. aphrodite* and the peloric mutants *Phalaenopsis* hyb. “Athens” and *Phalaenopsis* hyb. “Joy Fairy Tale”. All the plants were grown under natural light and temperature in the greenhouse of the Department of Biology (University of Naples Federico II, Napoli, Italy) or of the Department of Cell Biology and Plant Biochemistry (University of Regensburg, Regensburg, Germany).

The wild-type *Phalaenopsis* has the second floral whorl clearly distinguished into two lateral inner tepals and one median inner tepal (labellum or lip) (Figure 1a–c). Both peloric mutants have two labellum-like organs in substitution of the lateral inner tepals (Figure 1d,e).

Single flowers from three different plants of the wild-type *P. aphrodite* were collected before anthesis at different developmental stages: B1 (bud length 0.5–1 cm), B2 (1–1.5 cm), B3 (1.5–2 cm), B4 (2–2.5 cm), and B5 (2.5–3 cm) (Figure 1a,b). Open flowers (OFs) were collected soon after anthesis (Figure 1a,b). Single flowers of six wild-type *Phalaenopsis* hyb. “Athens” and of the peloric mutants were collected at developmental stage B2.

The perianth tissues (outer tepals, lateral inner tepals, and labellum) of all the collected flowers at the different developmental stages as well as the column and ovary were dissected and immediately frozen in liquid nitrogen or immersed in RNA*later* (Ambion, Austin, TX, USA) and stored at −80 °C until RNA extraction.

### 4.2. In Silico Identification of the PeDL1 and PeDL2 Genes

Total RNA was extracted from inner lateral tepals and labellum of wild-type and labellum-like lateral inner tepals from peloric *Phalaenopsis* hyb. “Athens” collected from 3 individual plants at the B2 developmental stage using Trizol (Ambion) followed by DNase treatment. After extraction, RNA was analyzed with the 2100 Bioanalyzer system (Agilent Technologies, Santa Clara, CA, USA) for sizing, quantitation, and quality control. Samples between 1 and 1.5 µg with an RIN (RNA integrity number) between 8.5 and 9.0 were sequenced by Macrogen (Seoul, Korea). Illumina TruSeq RNA (Oligo dT) mate paired-end libraries were generated and individually sequenced in a lane with a coverage >150 million 100 bp pair-end reads. For each sample 220 million paired-end reads were obtained. Analysis with FastQC showed that 94% of them had a quality score over 30. Trimming and mapping to the *Phalaenopsis equestris* genome v 1.0 (ASM126359v1) were carried out with the CLC Genomics Workbench (v11.01).

The Illumina raw reads of wild-type and peloric mutant *Phalaenopsis* hyb. “Brother Spring Dancer” KHM190 [59] were downloaded from the Sequence Read Archive. Paired-end reads from wild-type and peloric mutant outer tepal (accession numbers SRR1055198 and SRR1055947), inner tepal (SRR1055945 and SRR1055948), and labellum (SRR1055946 and SRR1055949) were assembled using the Trinity v2.3.0 software [60]. The Annocript v2.0.1 software [61] was used to obtain the functional annotation of the transcripts, and differential gene expression analysis between wild-type and peloric mutant tissues was performed with the edgeR v3.13 software [62].

The genomic organization of the *PeDL1* and *PeDL2* genes was reconstructed through BLAST analyses against the genome of *P. equestris* (assembly ASM126359v1), using as query the nucleotide sequence of the *DL*-like transcripts present in the transcriptome of *Phalaenopsis* hyb. “Brother Spring Dancer”.

### 4.3. Quantitative Expression Analysis

Total RNA was extracted from the tissues collected at the different developmental stages using Trizol (Ambion) followed by DNase treatment. After RNA extraction and quantification, equal amounts of total RNA were pooled, producing two pools for each tissue, each made of three different RNAs. Then, 500 ng of total RNA from each pool were reverse-transcribed using the Advantage RT-PCR kit (Clontech, Mountain View, CA, USA) and a mix of oligo dT and random hexamer primers.

The nucleotide sequences of the *PeDL1* and *PeDL2* transcripts and of their alternatively spliced isoforms identified by in silico analysis were verified through PCR amplification of the cDNA of *Phalaenopsis* hyb. “Athens” using gene- and isoform-specific primer pairs (Appendix A). The amplification products were cloned into pSC-A-amp/kan vector (Agilent Technologies, Santa Clara, CA, USA) and sequenced using the T3 and T7 primers (Eurofins Genomics, Ebersberg, Germany). The nucleotide sequences were deposited in GenBank with the following accession numbers: MW574592 (*PeDL1_1*), MW574593 (*PeDL1_2*), MW574594 (*PeDL2_1*), MW574595 (*PeDL2_2*).

Relative expression of *PeDL1* (two isoforms: *PeDL1_1* and *PeDL1_2*), *PeDL2* (two isoforms: *PeDL2_1* and *PeDL2_2*), *PeMADS2*, *PeMADS5*, *PeMADS3,* and *PeMADS4* was evaluated in all the collected tissues by qPCR experiments, using 18S, *Actin*, and *Elongation Factor 1α* as reference genes, as previously described [20]. The gene- and isoform-specific primer pairs used are listed in the Appendix A. At least one primer for each pair was constructed spanning two adjacent exons (Figure 2). The reactions were conducted in technical triplicates. Normalized relative quantity (NRQ) ± SEM was calculated for each replicate to the geometric average expression of three internal control genes [20].

ANOVA analysis followed by Holm–Sidak post-hoc test was performed to assess the statistical significance of the differences of NRQ among the different tissues.

### 4.4. Yeast Two-Hybrid Analysis

The GAL4-based yeast two-hybrid (Y2H) system (Matchmaker two-hybrid system; Clontech) was used to analyze protein–protein interactions between PeDL2_1 and PeMADS2-PeMADS6, and between the different isoforms of PeDL1 and PeDL2. As positive control, Y2H analysis was used to check the ability of PeMADS6 to form heterodimers with PeMADS2-PeMADS5.

The full-length coding regions of *PeDL1_1* (MW574592), PeDL1_2 (MW574593), PeDL2_1 (MW574594), PeDL2_2 (MW574595), *PeMADS2* (AY378149), *PeMADS3* (AY378150), *PeMADS4* (AY378147), *PeMADS5* (AY378148), and *PeMADS6* (AY678299) were amplified by PCR using the primer pairs listed in Appendix A and sub-cloned into the yeast expression vectors pGADT7 (prey) and pGBKT7 (bait) from the MATCHMAKER two-hybrid system 3 (Clontech), in frame with the sequence of either the transcription-activating (AD) or DNA-binding domains (BD) of the transcription factor GAL4. *Saccharomyces cerevisiae* strain AH109 was transformed with all the prey and bait recombinant vector combinations [63], conducting each experiment in triplicate.

Plasmid presence after double yeast transformations was verified by growing cells in Synthetic Defined (SD) medium lacking tryptophan and leucine. Protein–protein interaction was tested in SD medium lacking tryptophan, leucine, and histidine. Possible transcriptional activation activity of PeDLs and PeMADS2–6 proteins fused to the binding domain of GAL4 (pGBKT7 vector) was checked by monitoring the growth of yeast transformed cells in SD medium without histidine, in the presence of 20 mM 3-aminotriazole (3AT). Empty vectors pGBKT7 or pGADT7 were transformed in combination with the recombinant vectors as negative controls.

### 4.5. Identification of Conserved Motifs

Nucleotide sequences (3000 bp) upstream of the *PeDL2* gene of *P. equestris* were downloaded, as were the 3000 bp upstream of the *DcDL2* and *DcDL1* of *D. catenatum*. Unfortunately, it was impossible to download the sequence upstream of the *PeDL1* gene because the genomic scaffold starts with the coding sequence of this gene.

Putative promoter sequences were scanned for the presence of conserved motifs using the online tool MEME v5.3.3 [64] with the following parameters: motif width between 5 and 25, one occurrence of motif per sequence, and the maximum number of motifs 10. The search was repeated with the same parameters on the shuffled sequences as negative control. The identified conserved motifs were then checked against the JASPAR2020 Core Plants database (http://jaspar.genereg.net/, access date 18 January 2021) through TOMTOM v5.3.3 [65].

The search of known transcription factor binding sites within the putative promoters was conducted in PLANTPAN 3.0 [47]. In addition, using the FUZZNUC software (http://emboss.bioinformatics.nl/cgi-bin/emboss/fuzznuc), the putative promoters were scanned for the presence of perfect CArG-boxes CC(A/T)6GG and for the variants CC(A/T)7G and C(A/T)8G.

## Figures and Tables

**Figure 1 ijms-22-07025-f001:**
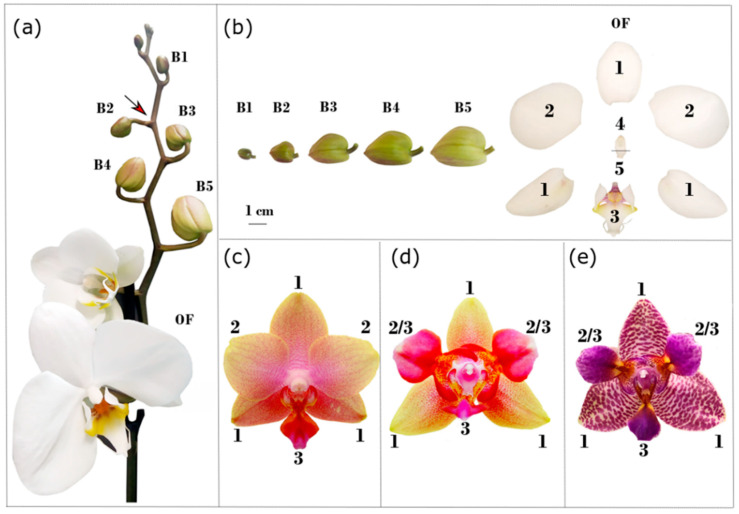
Wild-type and peloric mutants of *Phalaenopsis*. (**a**) Wild-type *P. aphrodite*; (**b**) floral buds at stages B1–B5 and floral organs at the OF stage of the wild-type *P. aphrodite*; (**c**) flower of the wild-type *Phalaenopsis* hyb. “Athens”; (**d**) flower of the peloric mutant *Phalaenopsis* hyb. “Athens”; (**e**) flower of the peloric mutant *Phalaenopsis* hyb. “Joy Fairy Tale”. The arrow in (a) indicates the point of rotation of the pedicel during resupination. Size of the developmental stages: B1 (0.5–1 cm), B2 (1–1.5 cm), B3 (1.5–2 cm), B4 (2–2.5 cm), B5 (2.5–3 cm), OF (open flower). 1, outer tepals; 2, lateral inner tepals; 3, labellum; 4, column; 5, ovary; 2/3, labellum-like organs.

**Figure 2 ijms-22-07025-f002:**
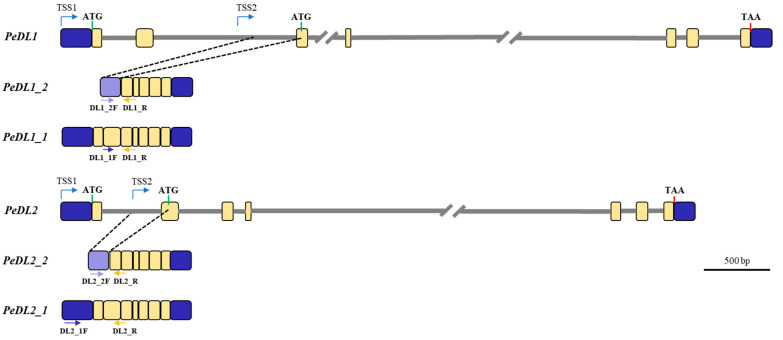
Genomic organization of the *PeDL1* and *PeDL2* transcribed regions of *P. equestris* and diagram of the corresponding alternative transcripts. The blue boxes represent the 5′- and 3′-UTRs; the yellow boxes represent the coding regions, the gray lines represent the introns. Introns of unknown size are shown as interrupted gray lines. The green and red bars indicate the position of the translation start (ATG) and stop (TAA) codons, respectively. TSS1 and 2 are the putative alternative transcription start sites of the different isoforms. The blue and yellow arrows indicate the position of the isoform-specific primer pairs.

**Figure 3 ijms-22-07025-f003:**
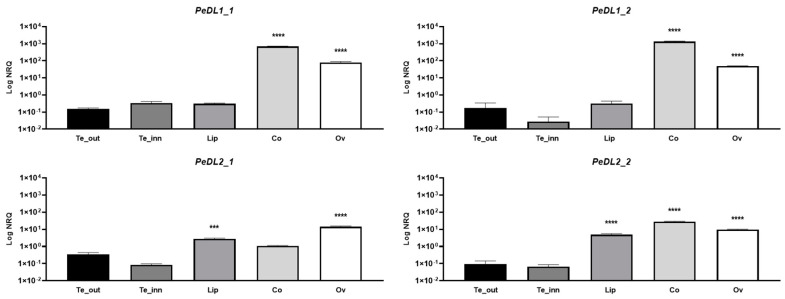
Relative expression of the different isoforms of the *PeDL1* and *PeDL2* genes in the floral tissues of the wild-type *Phalaenopsis* hyb. “Athens” at the B2 developmental stage (bud size 1–1.5 cm). The expression is reported as logarithm of the normalized relative quantity (Log NRQ). The bars represent the SEM of the biological and technical replicates. The asterisks indicate the statistically significant difference of the expression compared to outer tepals. *p*-Values *** <0.001, **** <0.0001. Te_out, outer tepals; Te_inn, lateral inner tepals; Co, column; Ov, ovary.

**Figure 4 ijms-22-07025-f004:**
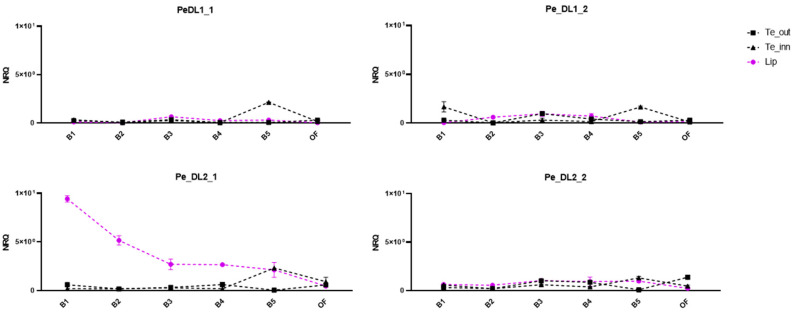
Relative expression of the different isoforms of the *PeDL1* and *PeDL2* genes in the perianth of the wild-type *P. aphrodite* at different developmental stages. The expression is reported as normalized relative quantity (NRQ). The bars represent the SEM of the biological and technical replicates. Bud size of the developmental stages: B1 (0.5–1 cm), B2 (1–1.5 cm), B3 (1.5–2 cm), B4 (2–2.5 cm), B5 (2.5–3 cm), OF (open flower). Te_out, outer tepals; Te_inn, lateral inner tepals.

**Figure 5 ijms-22-07025-f005:**
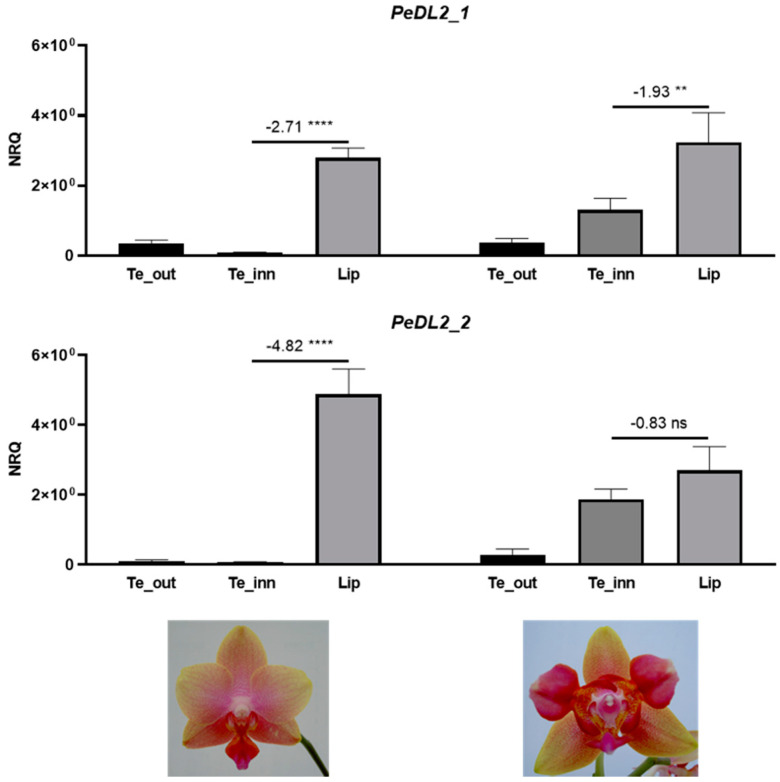
Relative expression of the isoforms *PeDL2_1* and *PeDL2_2* in the perianth of the wild-type (left) and peloric (right) *Phalaenopsis* hyb. “Athens” at the B2 developmental stage (bud size 1–1.5 cm). The expression is reported as normalized relative quantity (NRQ). The vertical bars represent the SEM of the biological and technical replicates. The numbers above the horizontal lines are the mean differences of the expression between lateral inner tepals and labellum (Te_inn - Lip). *p*-Values ** <0.01, **** <0.0001; ns, not significant. Te_out, outer tepals; Te_inn, lateral inner tepals or labellum-like structures that substitute the lateral inner tepals in the peloric mutant.

**Figure 6 ijms-22-07025-f006:**
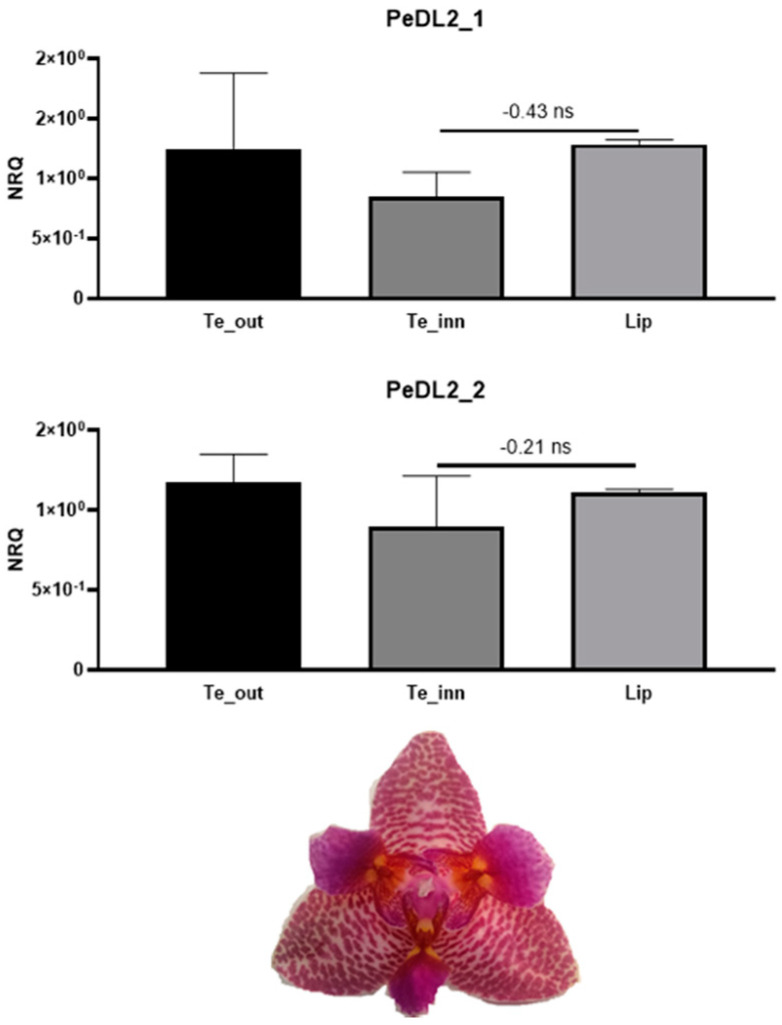
Relative expression of the isoforms *PeDL2_1* and *PeDL2_2* in the perianth of the peloric *Phalaenopsis* hyb. “Joy Fairy Tale” at the B2 developmental stage (bud size 1–1.5 cm). The expression is reported as normalized relative quantity (NRQ). The vertical bars represent the SEMs of the biological and technical replicates. The numbers above the horizontal lines are the mean differences of the expression between labellum-like structures and labellum (Te_inn-Lip). ns, not significant. Te_out, outer tepals; Te_inn, labellum-like structures that substitute the lateral inner tepals in the peloric mutant.

**Figure 7 ijms-22-07025-f007:**
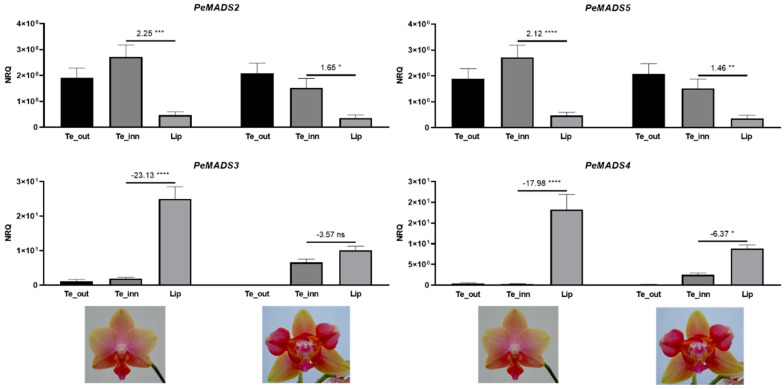
Relative expression of the class B MADS-box genes *PeMADS2-PeMADS5* in the perianth of *Phalaenopsis* hyb. “Athens” wild-type (left) and peloric mutant (right) at the B2 developmental stage (bud size 1–1.5 cm). The expression is reported as normalized relative quantity (NRQ). The vertical bars represent the SEM of the biological and technical replicates. The numbers above the horizontal lines are the mean differences of the expression between lateral inner tepals and labellum (Te_inn-Lip). *p*-Values * <0.05, ** <0.01, *** <0.001, **** <0.0001; ns, not significant. Te_out, outer tepals; Te_inn, lateral inner tepals or labellum-like structures that substitute the lateral inner tepals in the peloric mutant.

**Figure 8 ijms-22-07025-f008:**
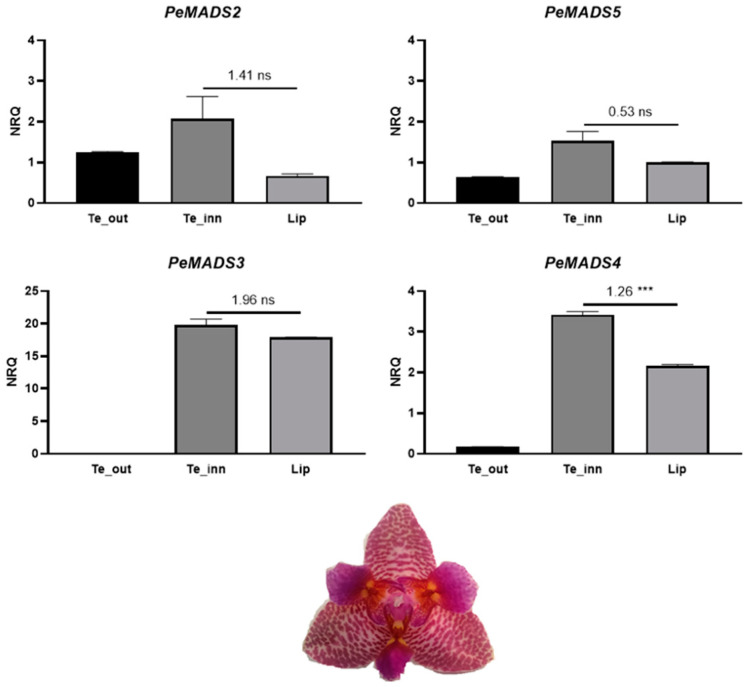
Relative expression of the class B MADS-box genes *PeMADS2–5* in the perianth of *Phalaenopsis* hyb. “Joy Fairy Tale” at the B2 developmental stage (bud size 1–1.5 cm). The expression is reported as normalized relative quantity (NRQ). The vertical bars represent the SEM of the biological and technical replicates. The numbers above the horizontal lines are the mean differences of the expression between lateral inner tepals and labellum-like structures (Te_inn-Lip). *p*-Values *** <0.001; ns, not significant. Te_out, outer tepals; Te_inn, labellum-like structures that substitute the lateral inner tepals in the peloric mutant. Note the different scale for *PeMADS3*.

**Figure 9 ijms-22-07025-f009:**
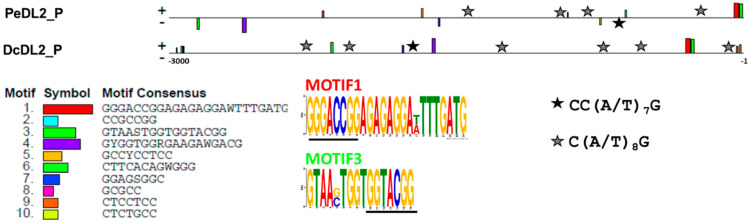
Conserved motifs within the putative promoters of the *DL2* genes of *P. equestris* and *D. catenatum*. PeDL2_P and DcDL2_P are the nucleotide sequences spanning 3000 bp upstream of the ATG translation start site, numbered from −1 to −3000. In the sequence logo of Motifs 1 and 3, the predicted binding site of the TCP factor (JASPAR IDs MA1096.1 and MA1035.1) and of the SBP-type zinc finger (JASPAR ID MA0955.1) are underlined. The black and gray stars indicate the CArG-box variants CC(A/T)_7_G and C(A/T)_8_G, respectively.

**Figure 10 ijms-22-07025-f010:**
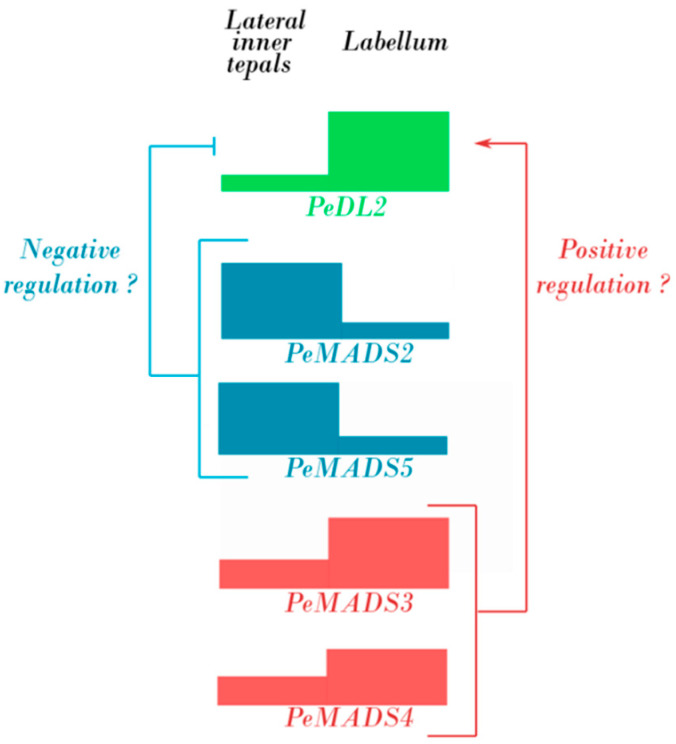
Possible regulatory interaction between PeDL2 and PeMADS2-PeMADS5 during the formation of the labellum of wild-type *Phalaenopsis*.

## Data Availability

The data presented in this study are openly available in NCBI (https://www.ncbi.nlm.nih.gov/) with the following accession numbers: PeDL1_1 (MW574592), PeDL1_2 (MW574593), PeDL2_1 (MW574594), PeDL2_2 (MW574595), PeMADS2 (AY378149), PeMADS3 (AY378150), PeMADS4 (AY378147), PeMADS5 (AY378148), PeMADS6 (AY678299), Phalaenopsis equestris genome v 1.0 (ASM126359v1), reads from wild-type and peloric mutant outer tepal (accession numbers SRR1055198 and SRR1055947), Phalaenopsis hyb. “Brother Spring Dancer” KHM190 Illumina reads of inner tepal (SRR1055945, wild-type, and SRR1055948, peloric), and labellum (SRR1055946, wild-type, and SRR1055949, peloric).

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
