# Peer review of "Extending the Toolkit for Beauty: Differential Co-Expression of DROOPING LEAF-Like and Class B MADS-Box Genes during Phalaenopsis Flower Development"

_ijms, 2021, doi:10.3390/ijms22137025_

Round 1
Reviewer 1 Report
Dear authors,
This article describes the expression pattern analysis of several genes in different parts of the Phalaenopsis flower. The experimental design seems logical and adequate , and the presented results look coherent. The manuscript is well written and methodology and results properly detailed, however, there are a couple of points that need to be improved:
In my opinion the discussion section is lacking some concluding remarks and possible future perspective considering their obtained results and how that is related what the authors state in the introduction not only about the proposed model but on their own proposal.
Although not a bad practice intrinsically, having 14 references out of 62 from the participating authors could give the impression of a narrower perspective, specially when there are other publications that have been left out that in my opinion are very relevant to this subject and thus a deeper literature review should be conducted. Some examples are:
Characterization of the Possible Roles for B Class MADS Box Genes in Regulation of Perianth Formation in Orchid. Chang et al., 2010.
A Modified ABCDE Model of Flowering in Orchids Based on Gene Expression Profiling Studies of the Moth Orchid Phalaenopsis Aphrodite. Su et al., 2013
Transcription analysis of peloric mutants of Phalaenopsis orchids derived from tissue culture. Chen et al. 2005.
PhalDB: A comprehensive database for molecular mining of the Phalaenopsis genome, transcriptome and miRNome. Lee et al., 2018
Author Response
Dear reviewer,
We are deeply grateful for your comments and suggestions.
We added some concluding remarks and possible future perspectives, as requested. In addition, we cited the suggested additional literature:
- Characterization of the Possible Roles for B Class MADS Box Genes in Regulation of Perianth Formation in Orchid. Chang et al., 2010.
- A Modified ABCDE Model of Flowering in Orchids Based on Gene Expression Profiling Studies of the Moth Orchid Phalaenopsis Aphrodite. Su et al., 2013
- Transcription analysis of peloric mutants of Phalaenopsis orchids derived from tissue culture. Chen et al. 2005.
- PhalDB: A comprehensive database for molecular mining of the Phalaenopsis genome, transcriptome and miRNome. Lee et al., 2018
We hope that the manuscript is now suitable for publication in the IJMS.
Best regards,
Serena Aceto
Mariana Mondragon-Palomino
Reviewer 2 Report
The research topic is of interest, and the work is based on an original hypothesis. On the other hand, the methods are not novel for the community and should be described more carefully. For that reason, I suggest the following minor revisions:
- The "Materials and Methods" section needs to be revised with the versions or date of accession for each software described in lines 573-575 and 632-639.
- "Supplementary_data_1.xlsx" and "Supplementary_data_2.xlsx" tables need improved legends, with at least the specification of the abbreviations for the sample names, and this has to be consistent in the columns "Sample A" and" Sample B". The colour codes must be specified as well.
- The "Supplementary_Material.docx" has to be adapted for the IJMS journal, with an improved resolution for Supplementary Figure 1 and clear photos for Supplementary Figure 4.
A general question is why the authors have chosen the specific mutants to conduct this study. The morphologic differences are well described, but it is necessary to clarify their importance in the Introduction section and link it to the results to justify the choice.
Overall, the manuscript has interesting results and an appropriate discussion, and I recommend the publication after the minor revision.
Author Response
Dear reviewer,
We are deeply grateful for your comments and suggestions.
Minor revisions:
- The "Materials and Methods" section needs to be revised with the versions or date of accession for each software described in lines 573-575 and 632-639.
- Answer: we added the reference for the SRA reads and the version of the software Trinity, Annocript, EdgeR, MEME and TOMTOM.
- "Supplementary_data_1.xlsx" and "Supplementary_data_2.xlsx" tables need improved legends, with at least the specification of the abbreviations for the sample names, and this has to be consistent in the columns "Sample A" and" Sample B". The color codes must be specified as well.
- Answer: we modified the legends of the supplementary data tables.
- The "Supplementary_Material.docx" has to be adapted for the IJMS journal, with an improved resolution for Supplementary Figure 1 and clear photos for Supplementary Figure 4.
- Answer: we tried to improve the resolution of Supplementary Figures 1, 4 and 5.
A general question is why the authors have chosen the specific mutants to conduct this study. The morphologic differences are well described, but it is necessary to clarify their importance in the Introduction section and link it to the results to justify the choice.
Answer: in the Introduction section, we clarified the importance of the peloric mutants in our study.
We hope that the manuscript is now suitable for publication in the IJMS.
Best regards,
Serena Aceto
Mariana Mondragon-Palomino